# A Flexible Fog Computing Design for Low-Power Consumption and Low Latency Applications

**Markos Losada** [1,*], **Ainhoa Cortés** [1,2,*], **Andoni Irizar** [1,2], **Javier Cejudo** [1] and **Alejandro Pérez** [1]

[1] CEIT-Basque Research and Technology Alliance (BRTA), Manuel Lardizabal 15, 20018 San Sebastián, Spain; airizar@ceit.es (A.I.); jcejudo@ceit.es (J.C.); aperez@ceit.es (A.P.)

[2] Tecnun, Universidad de Navarra, Manuel Lardizabal 13, 20018 San Sebastián, Spain

[*] Correspondence: mlosada@ceit.es (M.L.); acortes@ceit.es (A.C.); Tel.: +34-943212800 (A.C.)

**Abstract:** In this paper, we propose a flexible Fog Computing architecture in which the main features are that it allows us to select among two different communication links (WiFi and LoRa) on the fly and offers a low-power solution, thanks to the applied power management strategies at hardware and firmware level. The proposed Fog Computing architecture is formed by sensor nodes and an Internet of Things (IoT) gateway. In the case of LoRa, we have the choice of implementing the LoRaWAN and Application servers on the cloud or on the IoT gateway, avoiding, in this case, to send data to the Cloud. Additionally, we have presented an specific setup and methodology with the aim of measuring the sensor node's power consumption and making sure there is a fair comparison between the different alternatives among the two selected wireless communication links by varying the duty cycle, the size of the payload, and the Spreading Factor (SF). This research work is in the scope of the STARPORTS Interconnecta Project, where we have deployed two sensor nodes in the offshore platform of PLOCAN, which communicate with the IoT gateway located in the PLOCAN premises. In this case, we have used LoRa communications due to the required large distance between the IoT gateway and the nodes in the offshore platform (in the range of kilometers). This deployment demonstrates that the proposed solution operates in a real environment and that it is a low-power and robust approach since it is sending data to the IoT gateway during more than one year and it continues working.

**Keywords:** harsh environment; fog computing; edge computing; cloud computing; IoT gateway; LoRa; WiFi; low power consumption; low latency; flexible; smart port

## 1. Introduction

At the beginning of the new millennium, the increase in users connected to the Internet forced companies to rethink the way they used the Internet to offer their services. The modern wireless communication systems, the infrastructures required by the Internet and the increasing demand for large volumes of data, provided the perfect conditions for Cloud Computing to prosper. Keeping with this trend, computing, control, and data storage has been centralized and moved to the cloud, as was stated years ago in Reference [1]. Cloud Computing allows the possibility of storing and processing data without the need for a specialized HW and/or SW, as long as you have an Internet connection.

Internet of Things (IoT) is a collection of computing devices (specifically things) interconnected through the Internet and intended to offer services aimed at all kinds of applications [2]. Currently, many electronic devices that are part of the IoT are data producers. It is not difficult to think that, a few years from now, that number of devices will be multiplied. By 2025, it is estimated that 30 billion devices will be connected to Internet using Low Power Wide Area (LPWA) networks and proprietary or cellular technologies [3,4]. In this case, the amount of data to be processed in the conventional cloud will make data processing inefficient or even unfeasible.

To alleviate this problem, the concept of Edge Computing emerged. As data is increasingly produced at the edge of the network, it is more efficient to process the data right there. This means that most of the generated data are not transmitted to the cloud, but they are processed at the edge of the network. Several implementations of the Edge Computing principle have been proposed in Reference [5–7], among others: Mobile Cloud Computing (MCC) [8], Cloudlet Computing (CC) [9], and Mobile Edge Computing (MEC) [10]. The different and multiple ways of implementing Edge Computing resulted in new perspectives on the Edge Computing paradigm; hence, the term Fog Computing appeared. Fog Computing represents a complete architecture that distributes resources horizontally and vertically between Cloud-to-Things. As such, it is not just a trivial extension of the cloud, but rather a new actor interacting with cloud and IoT to assist and enhance their interaction [11].

The difference between Fog Computing and Edge Computing is subtle. Furthermore, we have not found in the literature a clear definition to differentiate the Edge Computing term from the Fog Computing term. Due to this ambiguity, we present in this paper how we define these architectures. The main difference between them is where the computational power is located. In Fog Computing, the intelligence is at a node closer to the IoT device. That node can be called IoT gateway. This fits the definition in Reference [12], where Fog Computing is defined as a horizontal, system-level architecture that distributes computing, storage, control, and networking functions closer to the users along a cloud-to-thing continuum. However, in Edge Computing, the edge is the IoT device responsible for the data generation and processing [13], and it is connected to the cloud.

Fog Computing is intended to solve the typical problems of Cloud Computing, such as:

- Unpredictable end-to-end network latencies between the end user and the cloud. Hence, Fog Computing can achieve better time responses, which is important for real-time applications and services.
- Frequent use of Cloud infrastructures. Fog Computing reduces the number of connections with the cloud and, therefore, possible interruptions in the data flow.
- High bandwidth and high energy needs to cope with the intense data traffic. Fog Computing reduces the required bandwidth of the communications with the cloud in a network with a large number of nodes or data since the processing can be distributed at different levels: edge level, gateway level, and cloud level, reducing significantly the quantity of data to be sent to the cloud.

Additionally, Fog Computing offers some advantages with respect to Edge Computing, such as:

- Increasing the security and privacy with the creation of a pre-cloud link to protect the data.
- Reducing the resources at the node to execute complex processes.
- Increasing the autonomy of the edge nodes with a significant reduction of their power consumption.

To our best of knowledge, most of the previous works related to Fog Computing architectures are reviews, surveys, or analysis of the current state of the art [13,14]. Few of them are architectural proposals based on Fog Computing. Furthermore, some implementations presented in the literature propose generic architectures to integrate Fog Computing in IoT-based applications [15] or they present test-bed and simulation results in order to evaluate the viability of the proposals [16]. Reference [17] presents an intra-vehicle resource sharing model with the aim of getting low-latency cloud services. Their motivation is in line with this research. However, they focus on the framework using mobile communications based on 5G technology and not on a low-cost and low-power deployment. In any case, these implementations do not show results in a real use case.

Among the most common technologies used for IoT devices, we can highlight Low Power Wide Area (LPWA). These technologies offer their ability to deliver low-power connectivity to a large number of devices spreading across large geographic areas at

an unprecedented low cost. A LoRa-type LPWA network uses a gateway, which can be connected to the cloud. LoRa Tx/Rx has low power consumption and long range compared to other LPWAs. This technology can be operated on sub-gigahertz unlicensed radio bands. Furthermore, with these characteristics, its market penetration and its wide use in the industrial, educational, and amateur community make LPWA LoRa technology ideal for IoT [3,18]. However, the key goal of LoRa technology is to achieve long range with low power consumption and low cost, unlike other technologies that are more appropriate to achieve higher data rates, lower latency, and higher reliability. There are situations where complex processing is required, but the node does not have the necessary resources for that processing. Hence, more data must be sent to the fog or cloud, and a higher data rate communication will be required.

This paper proposes a flexible architecture for IoT based on Fog Computing using LoRa and WiFi communications. This architecture can be easily implemented in many IoT applications. Many of these benefits are inherent to an architecture based on Fog Computing, that is, the Internet connection, security, and privacy and the limitation of resources on the edge, which are characteristics of the own architecture. As well as exploiting the benefits of Fog Computing, the proposed architecture permits us to select the most appropriate wireless connection with the IoT gateway according to the data rate, the payload size, the required range, and the power consumption. Furthermore, the edge nodes contain different sensors with low and medium data sizes, and the data processing can be distributed between the sensor nodes and the IoT gateway as needed. The proposed sensor nodes are ultra-low power solutions, thanks to the power strategies applied to the SW and HW designs. To quantify the benefits of the proposed approach, this paper will present specific results, such as power consumption at the edge node and at the IoT gateway.

The paper is structured as follows. In Section 2, a review of the different IoT applications are introduced. Section 3 provides the description of the case study of this research work. Section 4 presents the implementation of the proposed Fog architecture. In Section 5, the test setup, the measurement methodology, and the experimental results are presented. Finally, the discussions and conclusions obtained from the experimental results are summarized in Section 6.

## 2. Related Work

As commented in the previous Section, Cloud Computing has played a huge role in IoT applications. Some of these applications have started to demand faster execution in their processes. Hence, the trend has been to take advantage of the capabilities for computing on the edge devices to process data and, among other benefits, reduce the amount of data to be transmitted. More recently, the idea of processing data between the edge and the cloud has reported new benefits, such as an increase in network security and energy-size saving at the end nodes, making possible the implementation of this philosophy for new applications.

Among the applications for IoT reviewed in the literature, we have found several examples in which the rapid response of the system is an essential requirement [19]; therefore, Edge Computing has been used. For example, the increase in security applications, such as fire control, face recognition, or traffic control, have caused video surveillance and video analysis systems to have grown tremendously in recent decades. The algorithms for video analysis require intensive processing and with added privacy, as shown in Reference [20]. In Reference [21], a classification and a review of current architectures for Edge Computing is made, and an experimental analysis is presented for the case of image processing in the field of video games. In this case, Edge Computing performance for a recurse-intensive application is evaluated through different scenarios. Edge Computing satisfies the necessary requirements for these applications to the detriment of the size and consumption of the end nodes. In Reference [17], a vehicular infrastructure model for 5G technology is proposed, taking into account compute intensive applications but providing some kind of cloud assistance and looking for low-latency cloud services. Ref. [7] is more focused

on smart cities services at the edge providing security, privacy and protection to exploit edge servers computational, and low network latency capabilities. In spite of the fact that these last approaches look for an optimization of the distribution of the computational complexity for the provided services, none of them are focused on the reduction of the power consumption and cost of the overall architecture.

On the opposite side, there are other IoT applications in which the sensor nodes measure different parameters, and they send the information with the minimum necessary processing. In these applications, the communication of the sensor nodes with the cloud is carried out through low-bandwidth and long-range communications. These nodes are designed to operate for long periods without the need to replace the battery, but they can only work for very low data sizes which permits very small duty cycles. Smart cities are a typical example of IoT in modern infrastructures, which allow, by means of a sensor networks deployment, precise measurements of resources, such as water, electricity, and gas. Architectures, such as those presented in Reference [22–24], have been implemented in these scenarios. However, as has been discussed, the HW implemented in the end nodes is not flexible enough to enlarge the data sizes to be transmitted or to carry out more complex processing. Reference [6] proposes a Reinforcement Learning framework for autonomous energy management focused on mobile user devices. The proposed architecture learns the power-related statistics of the devices, providing a computation environment on the fly but only taking into account the cost of resource usage. As mentioned in Section 1, none of these approaches show results in a real use case.

In summary, we have seen that there are time-sensitive applications with a high computational load, while, in other applications, the fundamental requirement is the low power consumption. For these applications, non-flexible hardware architectures have been designed, i.e., the more efficient in power consumption, the less powerful to process data, or vice versa. Furthermore, remote sensing has opened the doors to new ways of monitoring and controlling a multitude of still unexplored fields, which will lead to the development of new IoT applications; thus, they will demand more flexible architectures. To deal with this problem, new architectures, such as Reference [15,16], have emerged. In these architectures, the computational load has been transferred to an edge gateway near to the end nodes. This edge gateway communicates with an upper gateway by means of LoRa, and this, in turn, communicates with the cloud. A simple HW to achieve a small size and low power consumption characterizes the end nodes. These nodes communicate with the edge gateway through Bluetooth. Hence, when the application requires it, they can send large amounts of data at relatively high speeds. In addition, complex processing is possible at the gateway to respond to the application in the cloud. Thus, these architectures solve certain problems presented in the literature by exploiting the benefits offered by a Fog Computing architecture. However, their approaches are not flexible in terms of the deployment of the end nodes since the gateway must be very close to them. Moreover, their approaches have not been deployed in a real environment and a real use case, which is essential for a system validation to check the applicability and the robustness of this kind of solutions. Furthermore, both communication technologies will always be active at the same time in order to send data to the cloud. This fact entails a re-transmission of the data, thus yielding an extra consumption.

## 3. System Applicability

The approach presented in this paper provides an efficient architecture for a wide range of applications, such as Smart Buildings, Smart Factories, and Smart Ports. These applications have in common the necessity of deploying low-cost and low-power IoT devices for monitoring the environment and/or the infrastructure degradation using wireless communications to facilitate their own deployment. Our architecture fulfills these costs and power requirements. Furthermore, our solution is capable of changing the wireless link from LoRa to WiFi, and, vice versa, according to the conditions of the specific situation, we can have for one application. As an example, for the Smart Factory

application, the IoT gateway could be integrated in an Autonomous Guided Vehicle (AGV) to gather the environmental and degradation data. Hence, the AGV can be capable of getting closer to some sensor nodes, but other ones can be located far away. When the AGV is stopped at the expected control points, our system will be able to receive the data in both scenarios. On the other hand, this will not be the only criterion to select one communication link. The system will be capable of making that decision according to some internal configurations, such as the required data rate and the size of the payload, which will affect the power consumption. These configurations can vary during the operation since one application can have different stages.

*Case Study: Smart Ports*

This research work is focused on Smart Ports as the case study to prove the efficiency of the proposed architecture. A port is a complex and dynamic environment that includes various activities, such as transportation, logistic, fishing, maintenance, and rescue operations, as well as protection of its environmental impacts [25]. The sensoring needs in a port ranges from localization of goods, ships, and infrastructure vehicles (which requires a constant update) to monitor the state of the docks, bollards, cranes, and warehouses, where updatings can be made on a daily or weekly basis. Additionally, ports are subjected to emergency situations (under heavy storms, for example) when it would be desirable to have more frequent updates of the state of the port's infrastructure. Therefore, IoT devices can considerably improve and automate many of these activities to increase the safety and security, as well as reduce the operational delays, of the different processes. However, this scenario also imposes strong conditions in the hardware, software, and network architectures of the IoT deployment. From the end-node perspective, there are two important restrictions: power availability and network access. The first one means that power consumption in the end-node must be drastically reduced in order to last several years without replacing the battery. The second means that the end-node must have some flexibility to access the network infrastructure based on range, power consumption, and data payload.

One of the most important advantages of the proposed architecture is that this design offers a high flexibility in terms of deployment. Hence, this approach can be easily customized to achieve an efficient solution for different applications.

Each application and use case will have different requirements related to latency, privacy, communications coverage, and power consumption. Our approach permits to configure the network on the fly according to these application requirements. In order to do that, the nodes can select the wireless communications to be used (WiFi or LoRa) and can communicate with a local server avoiding the communication with the cloud, if needed. On the other hand, we are capable to increase the computational complexity of the IoT gateway with the aim of reducing the quantity of data sent to the cloud. This fact would reduce the latency, which is a really important factor for real-time applications.

The specific application we have analyzed in this work is the predictive maintenance of critical infrastructures of the port, which can be made by the proposed system remotely and in an unattended manner. Thus, the sensor nodes must be deployed accordingly to measure the structural health of the most critical components or structures identified within the port. The proposed sensor nodes will be able to measure essential parameters to predict the structural health of these structures, such as the temperature, structural movements, corrosion, and pressure. Furthermore, our system is prepared to automatically use the most appropriate wireless connection according to the quantity of data (the size of the payload), the data rate required, and the distance to be covered. Our flexible platform will allow us to embed the IoT gateway into a drone so that, if the drone is capable of flying towards some sensor nodes, and high data rates are required, the WiFi connection will be selected. However, when the sensor nodes are located far away from the drone and over a restricted area where the drone cannot access, LoRa communications will be chosen with the aim of covering large distances. On top of that, there will exist other scenarios and use

cases where the criteria to select the communications link will be the efficiency in power consumption, and, for those cases, a further analysis is needed.

Taking into account the requirements of this specific application, the benefits obtained by using our proposed architecture are listed below:

- The aim is to monitor the structure remotely over years. To do so, the sensor nodes, which are deployed and embedded on the structure, must operate using a battery. Our approach applies some low power strategies to comply with this requirement looking for an autonomy of years. On the one hand, the sensor node will stay active when measuring or sending wireless data. On the other hand, the system will manage the wireless communications to be used taking into account the power consumption.
- In the scope of a port, there are critical structures everywhere. Therefore, it is important to cover different distances wirelessly. Our approach can select between WiFi connection when the range is not so long (around 100 m) and LoRa connection when we need higher ranges (in the range of kilometers). Other factor to select the wireless communications is the quantity of data to be sent. Thus, the WiFi connection will be active when large quantity of data are required and/or high data rates are needed but always if we have WiFi coverage (short range). Otherwise, the proposed solution will select between WiFi and LoRa communications according to the power consumption as is evaluated in this paper (see Section 5).
- To predict accurately the state of the critical structures and be able to act in real-time, a reduction of the latency in the communications can be considered as an important benefit. The proposed architecture permits to embed a local server inside the IoT gateway with the corresponding reduction of the overall latency.

This research work has been supported by the STARPORTS Project as is explained in the Funding section. PLOCAN, one of the STARPORTS participants, has an offshore platform very suitable to validate the proposed architecture in a relevant environment but at the same time in a controlled scenario. The offshore platform is 5.7 km from the PLOCAN building, very well suited to test LoRa communications when the IoT gateway is in the PLOCAN premises. Thus, two sensor nodes were deployed in the PLOCAN offshore platform (see Figure 1), and a fixed IoT gateway was installed in the PLOCAN building receiving data through LoRa from the sensor nodes over more than twelve months, working with a Spreading Factor (SF) of 10, in this case (currently the sensor nodes continue working with the same battery keeping good and stable voltage levels).

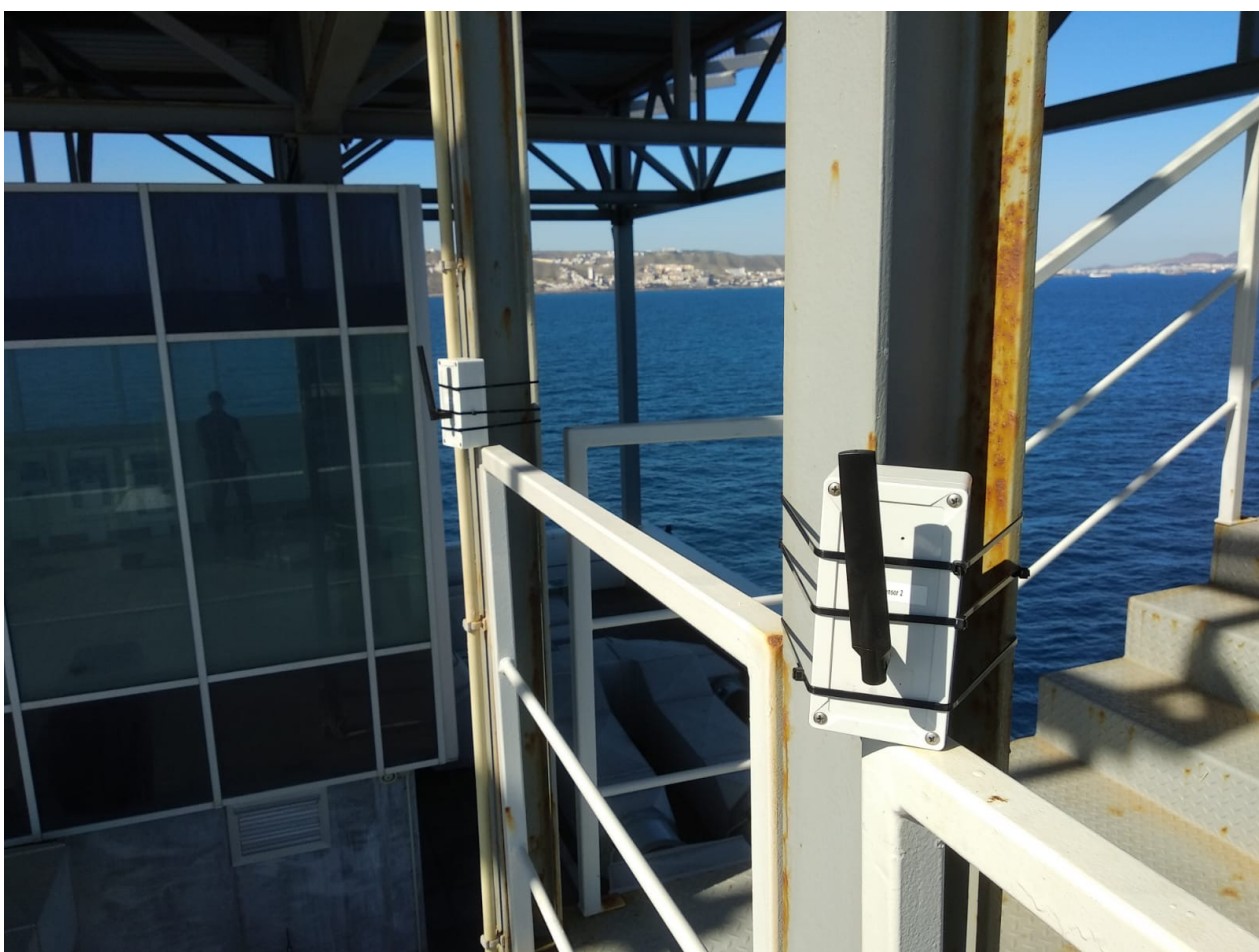

**Figure 1.** The sensor nodes deployed in the PLOCAN offshore platform.

## 4. Design of the Proposed Fog Architecture

The proposed architecture is shown in Figure 2. Note that the red boxes represent the deployed devices such as it is described in the previous Section. In this architecture, the IoT application located in the cloud communicates with the sensor nodes through an IoT gateway with two different wireless communication technologies, which are WiFi and LoRa. This gateway can be fixed or mobile, and it can communicate with each end-node through one of these communication technologies, depending on the demanded data rate, gateway to end-node coverage, or a trade-off between these parameters and power consumption. Cloud communication with the IoT gateway could be done through an Ethernet connection, in the case of a fixed IoT gateway, as in the case of the STARPORTS project, or a 3G/4G connection, in the case of a mobile IoT gateway. Additionally, the IoT gateway is capable of performing complex processes, such as filtering, calibration, correlation, frequency analysis, etc., to alleviate the data load in the cloud or executing safety-critical computation, if needed.

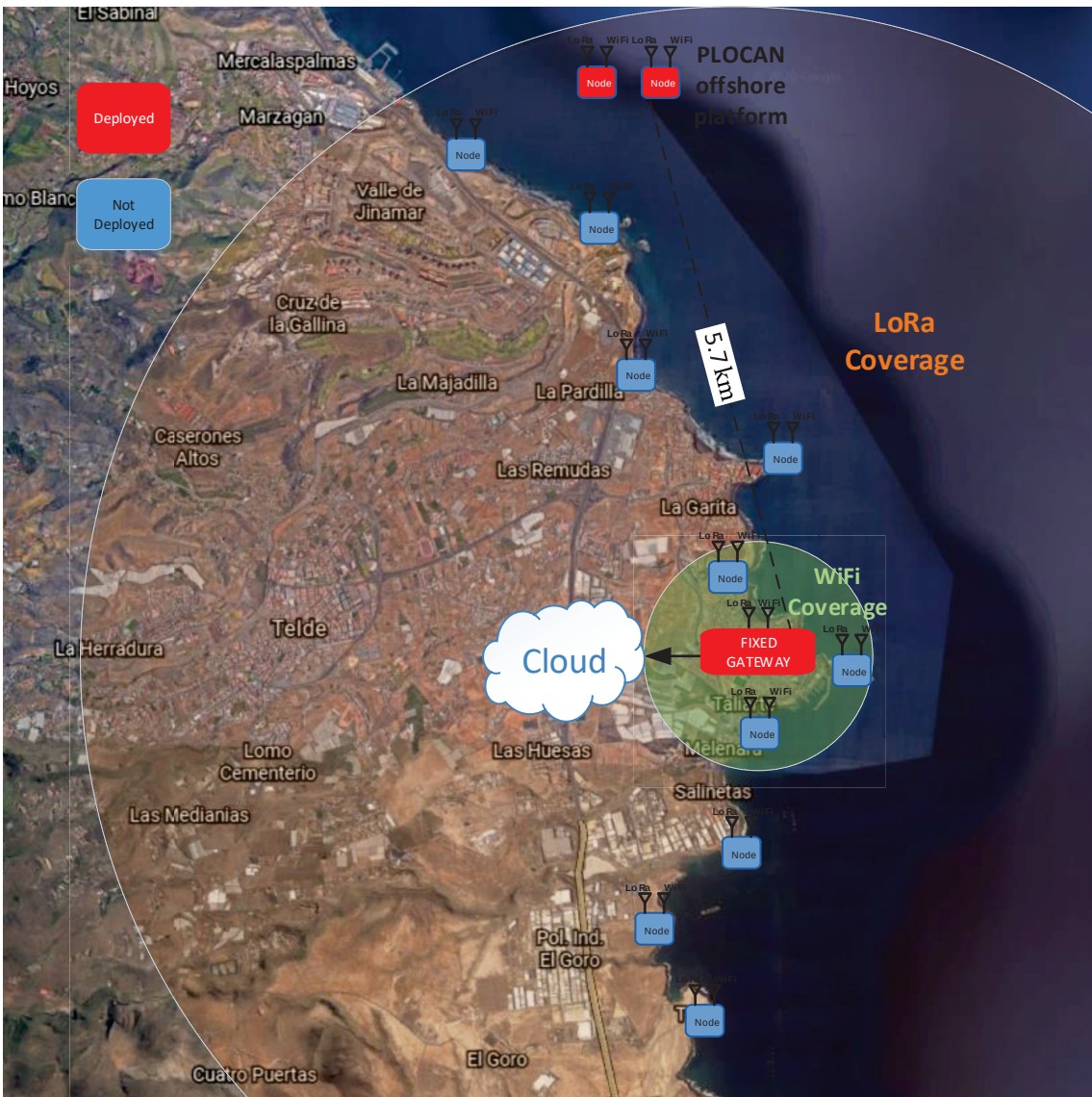

**Figure 2.** Proposed flexible Fog Computing architecture.

In this architecture, the sensor nodes can be considered low-cost and low-power nodes since an effective power management is applied and the proposed architecture permits not only to alleviate the data load in the cloud but also to distribute in an efficient manner the computational operations between the sensor nodes and the IoT gateway.

The total cost of the sensor node materials is shown in Table 1. In this case, accurate sensors have been used to measure acceleration, temperature, humidity, and corrosion. If the application does not require those measurements, the cost of the sensor node could fall below 30€.

The cost of the IoT gateway materials is shown in Table 2. In our prototype, the IoT gateway has been built using carrier boards. However, we could customize it by designing our own carrier board. In that case, it would only be necessary to acquire the SOM and the LoRa concentrator. Note that the cost of the SOM starts from 54€. The SOM can be customized as a function of the required features; therefore, this cost will depend on the SOM customization level.

**Table 1.** Total cost of the sensor node materials.

| Component | Part Name | Unit Price (€) | ×1000 (€) |
|---|---|---|---|
| LoRa Transceiver Module | RN2483 | 10.81 | 9.84 |
| Wireless Module | CC3220MODASF12MONR | 15.94 | 10.24 |
| Inductance-to-Digital Converter | LDC1000QPWRQ1 | 7.73 | 4.22 |
| Humidity and Pressure Sensor | BME280 | 5 | 2.14 |
| MEMS Accelerometer | ADXL355BEZ | 40.94 | 32.71 |
| Real Time Clock | DS1374U-18+ | 3.15 | 1.66 |
| Power management ICs | | <7 | <4 |
| Connectors (UFL, Battery connector) | | <2 | <1 |
| Other components | | <2 | <1 |
| PCB | Lab Circuits manufacturer | <10 | <5 |
| Total (€) | | <93 | <65 |

**Table 2.** Total cost of the Internet of Things (IoT) gateway.

| Component | Part Name | Unit Price (€) |
|---|---|---|
| SOM | DART-MX8M | Starting from 54 |
| SOM + Carrier Board | VAR-DT8MCustomBoard | 244 |
| LoRa Gateway Concentrator Module | RAK833 | 80 |

### 4.1. The Sensor Nodes

In a Smart Port environment, it would be desirable that the sensor nodes to be deployed work unattended with battery lives of more than 3 years [18]. Other important aspect is the communication range since in that environment you can need short and long range communications. Then, having such flexibility will permit large distances (several kms) but also small distances (<100 m) if a mobile platform is used to gather data from the sensor nodes (using a drone), assuming that the drone is stopped when the system gathers the data. Additionally, in a Smart Port you can have different kind of variables to be measured. Hence, the system must be able to measure from very low sample rate measures to high sample rate variables (∼500 Hz).

The architecture we have developed to fulfill the above features is given in Figure 3. Figure 4 shows a photograph of the sensor node electronics and housing. As can be seen, the node has wireless connectivity with WiFi and LoRa. The main component is the CC3220 micro-controller [26], a Cortex-M4 based device that includes a Network Processor with WiFi Driver, TCP/IP Stack, baseband processor, and complete analog front-end. It also contains a 1MB flash memory that allows storing data and configuration parameters in the node. LoRa connectivity is provided by the RN2483 module [27], which includes LoRaWAN Class A protocol stack). The CC3220 controls the LoRa device using commands via an UART interface.

The node is housing several sensors: 3-axis accelerometer (ADXL355 [28]) with very low noise density (22.5 μg/$\sqrt{\text{Hz}}$), a Pressure/Humidity/Temperature sensor (BME280 [29]) and an inductance sensor (LDC1000 [30]) that allows us to measure corrosion levels. All the three sensors can be accessed through a unique SPI interface.

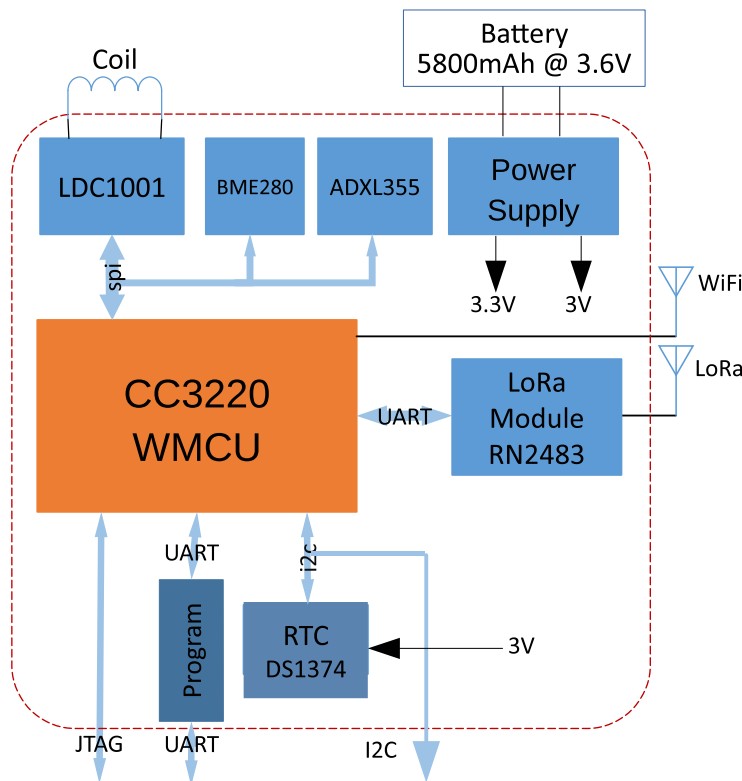

**Figure 3.** Block diagram of the sensor node.

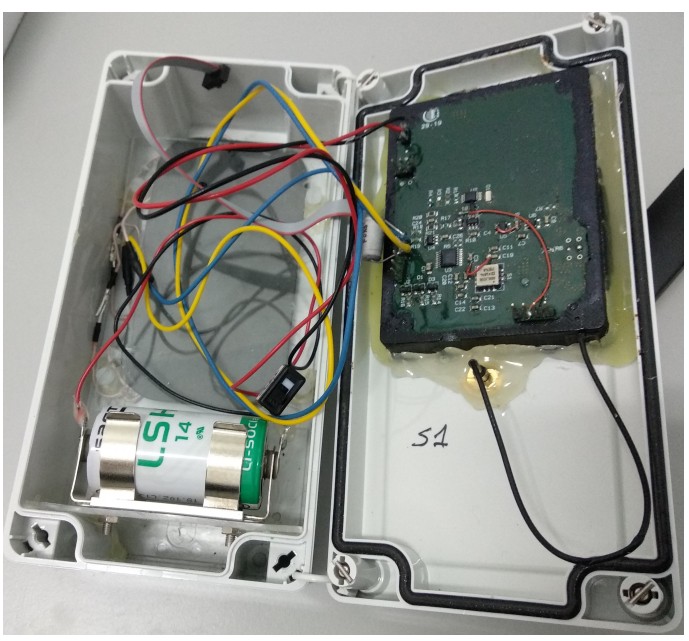

**Figure 4.** Sensor node.

Low Power Strategies

The power supply is provided by a primary battery of 3.6 V and 5800 mAh. For this battery size and given the required duration, this means the node must draw an average current of less than 250 µA. This value is too small, even for the lowest power micro-controller in the market (~135 µA/MHz.) Therefore, remote nodes must reduce its power consumption by decreasing its duty cycle. The duty cycle is the relationship between the on-time, the time in which the nodes are working, and the cycle time, which is the total time of one cycle. The off-time is the time in which the nodes are sleeping. The

node can modify the duty cycle by configuring the DS1374 RTC device [31] of Figure 3. The node will generate two supply voltages: 3 V for the RTC (see Figure 3) and 3.3 V for the rest of the circuit. The 3 V voltage is always active, but the 3.3 V can be enabled by a switch (first time boot) or by the RTC (generates a pulse that boots the CC3220 after a specified period according to the selected duty cycle). The 3.3 V will be disabled by the CC3220 after all the tasks related to a set of measures. Additionally, each sensor can be individually shutdown from the CC3220 using GPIO signals.

The sensor node has two operational modes: Normal Mode and Storage Mode, for which state machines are given in Figure 5. In Normal mode, the node typically powers up from a RTC pulse (POWER-UP state), loads design parameters from an internal non-volatile memory, takes measures from the sensors (SENSOR state), configures the wireless interface, and sends the data to the IoT gateway (LoRa-WIFI state). The node then waits for the Application server to send new configuration parameters for the next cycle (REMOTE state) and then goes back to the DEEP-SLEEP state by disabling the 3.3 V regulator output. In Storage Mode, the node stores the measures from the sensors in a non-volatile memory and only sends the complete data packet once every several cycles. This Storage Mode can be interesting in the case where we use WiFi communications from a mobile platform, such as a drone. Hence, the sensor node would store the data from areas without WiFi coverage until getting close to the IoT gateway integrated in the drone. On the other hand, both in the case of a fixed and a mobile approach when we need a real-time monitoring for fast processes, the Normal Mode is more suitable.

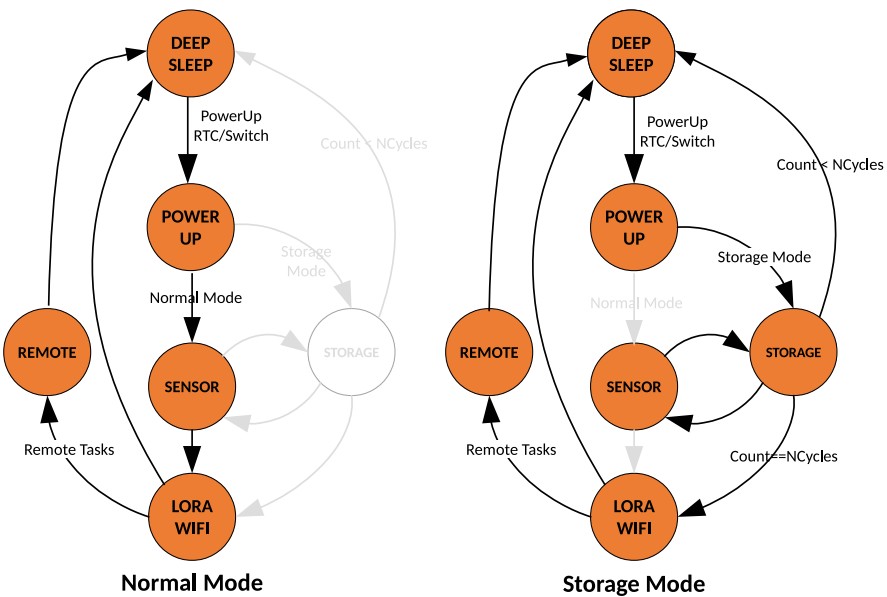

**Figure 5.** Operational modes and state machine of the sensor node.

LoRa and WiFi are two very different communications systems that complement each other very well in several aspects of a wireless link. Range, bit-rate, and power are the main design variables to consider when selecting a wireless interface. When a decision must be made based on range and bit-rate, the choice can be quite easy. But decisions based on power can be much more difficult because one must consider not only the physical layer of the communication interface but also the upper software layers. For example, from the strict specification of the modulation schemes, WiFi is much more power efficient than LoRa (in terms of nJ/bit, it is around three orders of magnitude more efficient). But, if we take into account the overhead introduced by the protocols and software stacks needed to manage the communications with both methods (much more complex in WiFi than in LoRa), the differences begin to narrow.

Low Power strategies must weight the wireless connection to be used (WiFi or LoRa) depending on the size of the payload, the duty cycle of the sensor node, and the commu-

nication range. This is the aim of the experiments carried out in this research work are described in detail in Section 5.

*4.2. The IoT Gateway*

A block diagram of the IoT gateway is presented in Figure 6. The IoT gateway designed is based on a Variscite DART-MX8M System-on-Module (SOM [32]) based on NXP's i.MX8M with up to 1.8 GHz Quad-core ARM Cortex-A53™, plus 400 MHz Cortex-M4™ real-time processor [33], and works under Debian (stretch) GNU/Linux 9. Although the DART-MX8M contains extensive processing capabilities from its quad-core architecture plus graphics and video processing unit, this SOM is not well suited for running Machine Learning or Artificial Intelligence applications in the gateway. However, our architecture can be easily upgraded to house the DART-MX8M-PLUS, pin-to-pin compatible with the DART-MX8M. The DART-MX8M-PLUS [34] includes the iMX8M-Plus processor [35] that has basically same quad-core Cortex-A53 architecture, plus a Cortex-M7 at 800 MHz and a Digital Signal Processor (DSP) accelerator also at 800 MHz. But, more importantly, it includes a Neural Processing Unit (NPU) that allow the efficient implementation of Machine Learning algorithms in the IoT gateway, reducing power and time consuming data transfers to the cloud.

To include the LoRa connectivity to our IoT gateway we have added a LoRaWAN concentrator [36] that interfaces with the processor through a SPI port. The software that manages the incoming LoRa packets from the concentrator is the Packet Forwarder from Semtech [37]. This software sends the encrypted LoRa packets to the LoRaWAN server of your choice, where they are decrypted by the LoRaWAN and Application servers. For this configuration, we have used The Things Network [38], as is shown in Figure 7. The IoT gateway will also be working as a WiFi Access Point. A software running in the SOM will be responsible for acquiring the data packets from the node.

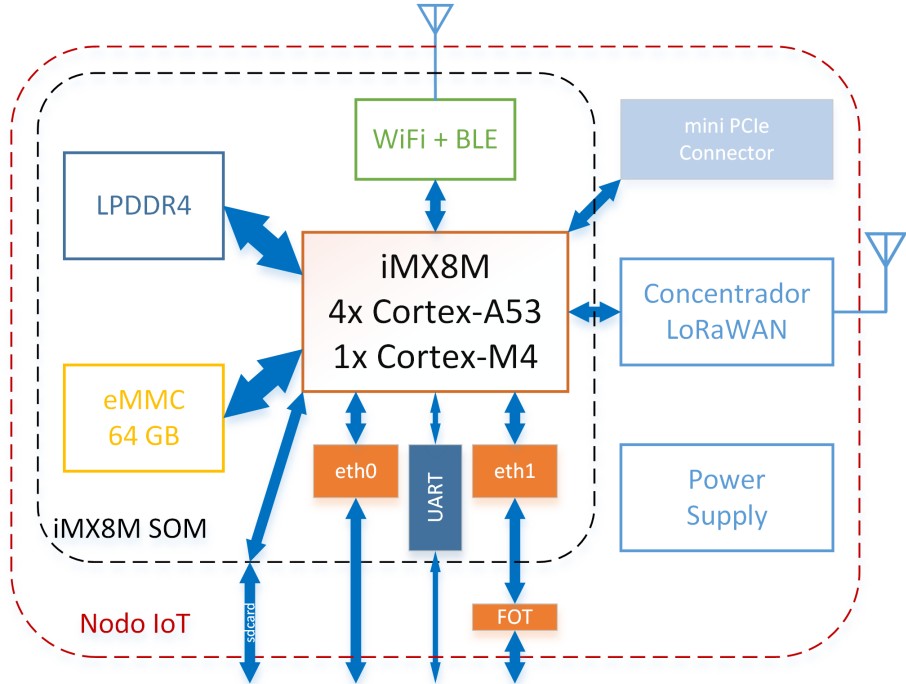

**Figure 6.** Block diagram of the IoT gateway.

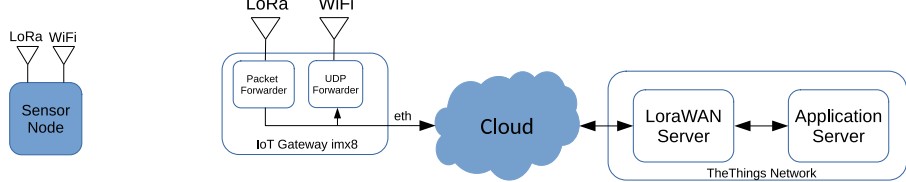

**Figure 7.** The IoT gateway with the server implemented on the cloud.

The IoT gateway has also the possibility to work, as shown in Figure 8, by installing both the LoRaWAN server and the Application servers in the i.MX8M SOM. As a result, data from the sensors can be processed, normalized, enhanced, and stored at the gateway, reducing the uplink traffic to the cloud. Figure 8 also shows the software architecture of the IoT gateway, an adaptation to the specifications of OpenFog reference architecture [12]. In our case, the physical layers are WiFi and LoRa, and the Protocol Abstraction layers are implemented, in this case, as packet forwarders that respond to the central component of the architecture, i.e., the Node Configuration and Management block. This block has access to the processing and storage resources of the IoT gateway and implements the communications with the cloud infrastructure.

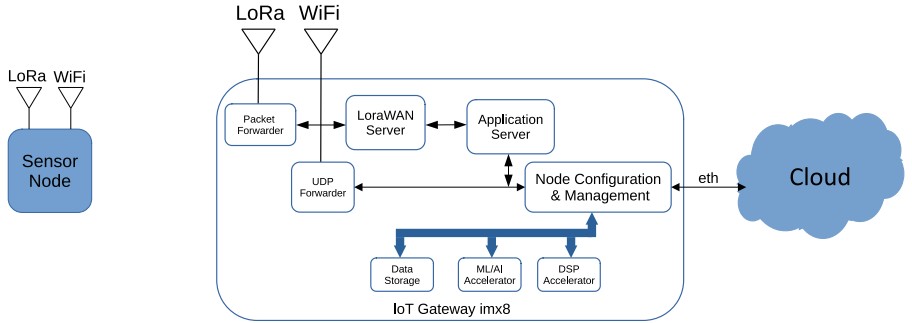

**Figure 8.** The IoT gateway software architecture with the LoRaWAN server implemented.

## 5. Results

### 5.1. Setup and Methodology

The aim of the proposed setup is to measure the energy consumption generated by the sensor node when using WiFi communication or LoRaWAN communication. In order to do that, we have implemented the setup shown in Figure 9. The same setup configuration has been implemented for the both cases, WiFi and LoRa, to carry out parallel tests. The power supply has been set to 3.3 V. The LTC4150 Coulomb counter monitors the current through a precision external resistor between the positive terminal of the source and the power terminal of the sensor node.

A series of pulses are obtained at the output of the Coulomb counter depending on the current consumed by both, the sensor node, and the Coulomb counter. Each pulse corresponds to a quantity of electric charge of 0.307 Coulombs, that is, 0.085 mA/h. A msp430FR2433 [39] micro-controller is responsible for capturing the instant of time in which each pulse was generated. The micro-controller sends the time instant and the number of the pulse to a PC via UART communication for further processing. Note that the micro-controller is powered by the computer; therefore, the consumption generated by itself is not taken into account.

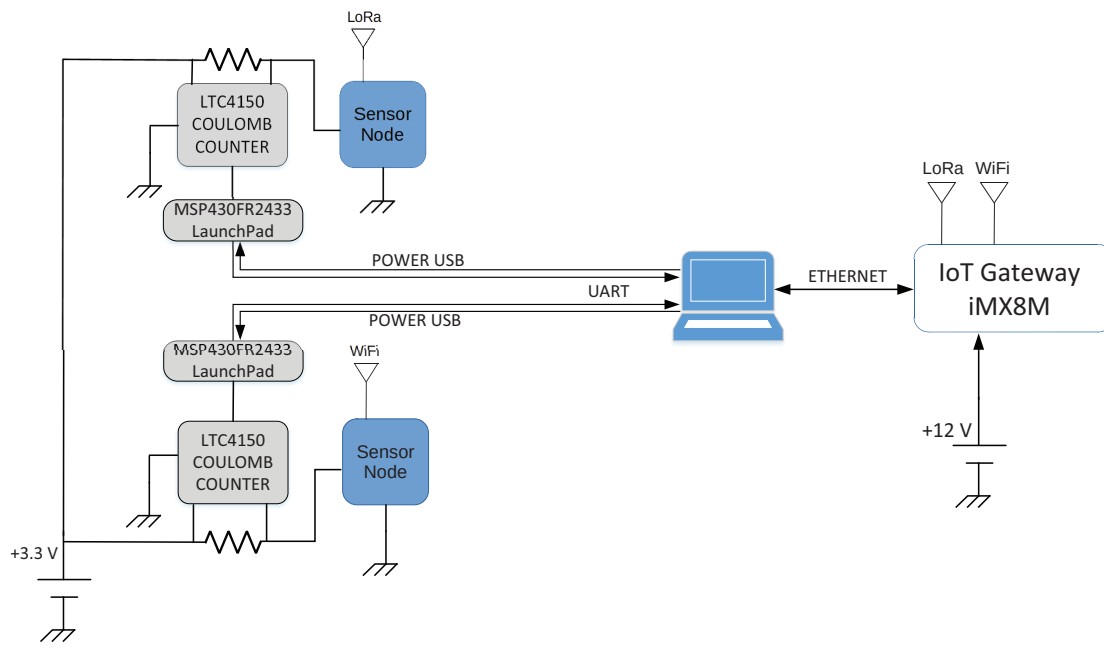

**Figure 9.** Implemented setup for the energy consumption measurements.

As can be seen in the photogragh of Figure 10, the msp430FR2433 LaunchPad is powered from the PC through the USB connector, whereas the power supply is used to power the sensor nodes and the Coulomb counter at 3.3 V. The LoRa and WiFi gateway have been both implemented in the same Variscite DART-MX8M SOM. A motherboard is used to connect and power the SOM.

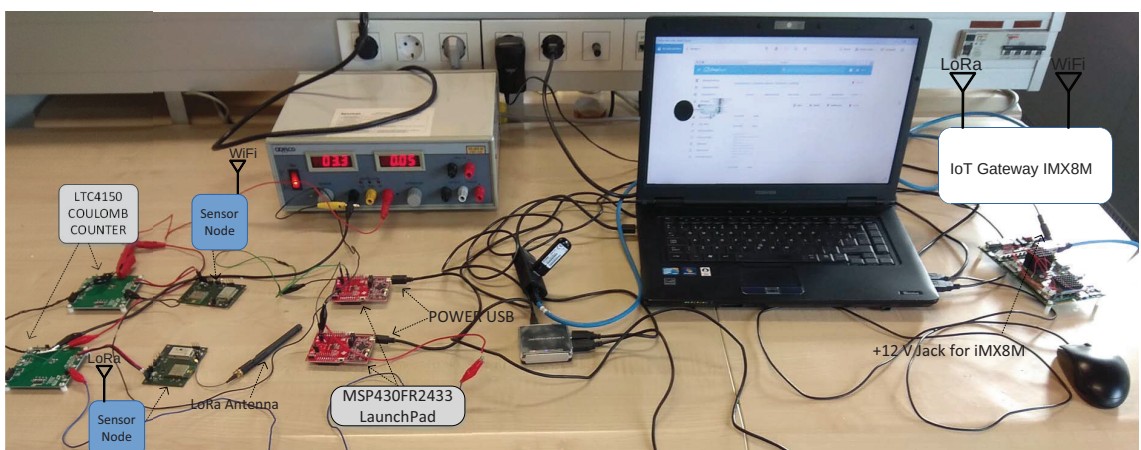

**Figure 10.** Photogragh of the implemented setup.

### 5.1.1. Lora and Wifi Gateway Implementation for the Energy Consumption Measurements

- WiFi gateway configuration:
  The WiFi module of the SOM has been configured as an access point, with static IP address and wpa-psk security type. Therefore, the node just needs to connect to the Wifi network generated by the SOM and send its frames.
  In order to receive and manage the received frames, a UDP socket to the WiFi IP address has been created so that every received frame is captured and logged into a file for further analysis.
- LoRa gateway configuration:

As explained in Section 4.2, a RAK833 LoRa concentrator has been used running "LoRa Packet Forwarder". The measured consumption of this LoRa concentrator is 420 mA at 3.3 V. The "LoRa Packet Forwarder" runs on the SOM to forward LoRa packets received by the concentrator to a server through an IP/UDP link. As is shown in Figures 7 and 8, the LoraWAN Server and the Application server can be located on the gateway or on the cloud. For the energy consumption measurements, we have used the configuration mode with the server on the IoT gateway to facilitate the overall setup and the data analysis since The Things Network has several restrictions related to higher data rates and payload sizes according to the SF. That is why we have implemented a Network server stack (Chirpstack) [40] into the SOM to carry out the tests. The Chirpstack LoRa Network Server stack provides open-source components for LoRa networks. Together, they form a ready-to-use solution including an user-friendly web-interface for device management and APIs for integration. The measured consumption of the SOM when Chirpstack is running is 800 mA, supplied at 3.3 V.

### 5.1.2. Sensor Node Implementation for the Energy Consumption Measurements

The operation of both tested sensor nodes is exactly the same, except that, in one case, it is used WiFi, and, in the other case, LoRa, but both nodes measure, process, and transmit the same number of data, with the same duty cycle and using the same operating modes for further comparison.

To check what communication is more efficient depending on the conditions, different tests have been done with different payloads for every Spreading Factor (SF). Note that different SFs affect mainly the range of LoRa communications.

The software running in both nodes follows the flowchart shown in Figure 11. This flow chart is in line with the normal mode described in Figure 5. As the on-time is not a fixed value due to the communication link, the on-time of each cycle must be measured. Thus, when the node starts a new cycle, it initiates a timer to count the on-time period of the duty cycle. Then, when the serial ports configuration is done, the configuration parameters stored in the flash memory files are read. These parameters are shown in Table 3. After reading the files, the node sets the cycle time in the DS1374 Real Time Clock. Then, the node is configured to use LoRa or WiFi communication based on the mode parameter.

If LoRa mode is active, the node initiates UART interface to communicate with RN2483. The LoRa module is configured with "automatic re-transmit" option and with the SF selected for the test. The frequency plan used is EU863-870. Once the LoRa module configuration is done, the node connects to LoRa network using Activation by Personalization (ABP). The main advantage of this type of connection is that it is not required to join the network in order to send data, that is, the server-side confirmation is not necessary since the session is already manually assigned. It has been preferred over Over the Air Activation (OTAA) because OTAA needs the LoRa module to be active all the time to have the session parameters updated. This means additional power is needed during the end-node's off-time, which can distort the power measurements in our experimental setup.

If WiFi mode is selected, the WiFi network processor subsystem is configured in station mode, with Dynamic Host Configuration Protocol (DHCP), normal power management policy, and auto connection policy. The WiFi standard used is 802.11 b/g/n 2.4 GHz. Once this network processor is configured, the node sets the WiFi network Service Set Identifier (SSID) and connects to it.

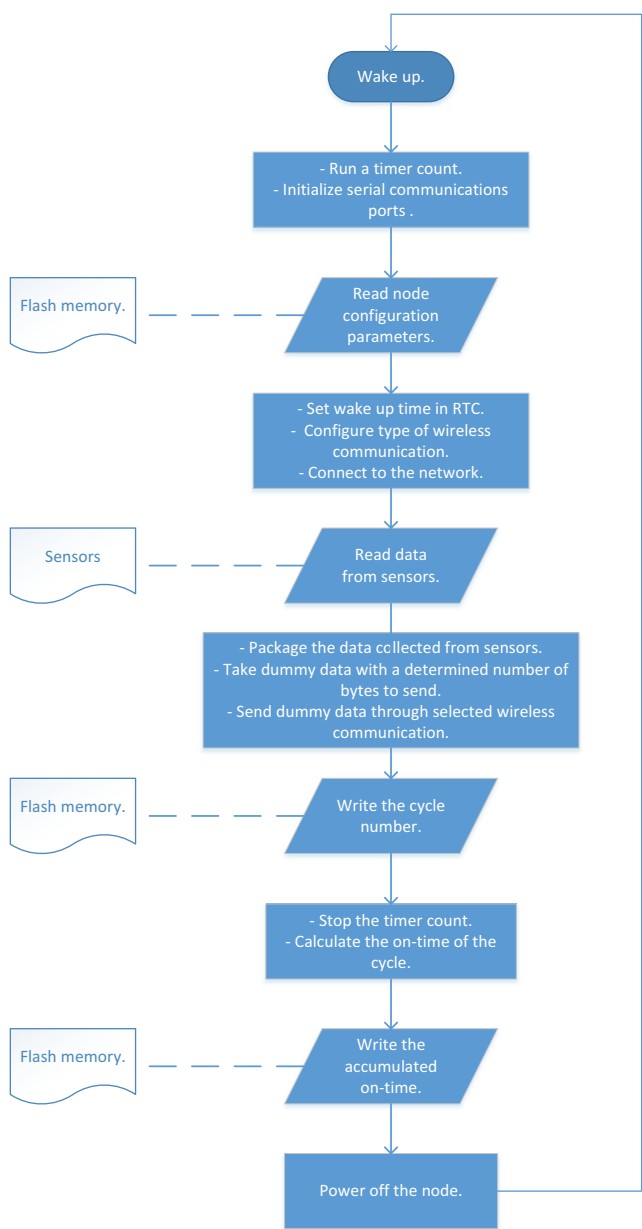

**Figure 11.** Sensor node software flowchart.

**Table 3.** Configuration parameters from the flash memory.

| Parameter | Description | Value |
|---|---|---|
| mode | Selects WiFi or LoRaWAN communication | Variable for each test |
| ssid | SSID of the WiFi network | STARPORTS |
| cycle time | The total time of one cycle | 20 s |
| payload | Number of bytes to be sent | Variable for each test |
| cycles | Number of cycles of the test | 4000 |

When the connection is established, either with the LoRa network or with the WiFi network, the node starts reading data from sensors. The first sensor is an Analog to Digital Converter used to calculate the voltage from battery. The second sensor is ADXL355 accelerometer. After this sensor configuration, the node reads the data from the accelerometer and packages it. The last sensor is BME280, a combined digital humidity, pressure, and temperature sensor. Before getting the data from BME280, some steps are carried out, such as calibrating sensors and compensating measurements. Then, the data obtained from BME280 are packaged.

In order to measure the energy consumption of our approach during the proposed tests, we decided to use the same payload size in each test to compare fairly the consumption results obtained with WiFi and LoRa links. Therefore, the node discards the real data from sensors and takes dummy data with a fixed number of bytes. This fixed payload is configured by reading a payload file from the flash memory according to the payload parameter, described in Table 3, which will select the payload file to be read. The node sends the packet over WiFi or LoRa network depending on the mode configured on the node. Hence, although the data from the sensors are not really sent wirelessly, the measurements done by these sensors have been configured in the same way for all the tests. Thus, we will have the same consumption according to the sensors for each test.

Finally, the cycle number is increased by one and written to a file stored in the flash memory. The timer counter stops and the accumulated on-time of the test is calculated by adding to it the on-time of the cycle, which is written in the internal flash memory. Then, the node goes to the sleep mode until the wake-up time signal from the RTC is received. Once the node completes the total number of cycles as is indicated in Table 3, the test finalizes.

*5.2. Energy Consumption Measurements*

Figure 12 shows the energy consumed by the sensor node in a 20 s cycle for different Spreading Factors (SF). Note that the SF does not apply to the WiFi technology. Despite this fact, we have evaluated WiFi and LoRa links with parallel tests done at the same time. Hence, as the SF affects LoRa, we have also repeated the WiFi tests for each SF.

In the case of LoRa, it can be seen that the energy consumption increases as the number of bytes to be transmitted or the SF is increased. However, in the case of WiFi, it can be seen that the energy consumption is more stable for different sizes of the payload, that is, the number of data transmitted does not have a great influence on the energy consumption. This effect is closely related to the duty cycle. In the case of LoRa technology, the time the data is at the air is longer than in the case of WiFi technology. Hence, the duty cycle is larger when LoRa is used, which implies a higher energy consumption when we have more bytes or higher SFs. Finally, it can be said that there is a cut-off point between the LoRa and WiFi curves, which, from an energy point of view and depending on the range we need to reach, can help us to determine which technology is more efficient. In this case, we see that, for SF's greater than 10, WiFi technology always consumes less, and, for an SF of 7, LoRa consumes less. For the case of an SF equal to 8, LoRa consumes more than WiFi when more than 140 bytes are transmitted, and, for the case of an SF equal to 9, LoRa consumes more than WiFi when we need to transmit more than 40 bytes.

From the tests carried out, the results have been extrapolated to determine the battery life as a function of the cycle time. Figure 13 shows the duration of a battery with a capacity of 5800 mA/h for different cycle times, different SFs, and different sizes of messages to be transmitted. It can be seen that the cycle time is the parameter that mostly determines the battery life, that is, for a specific SF configuration and data size to be transmitted, the longer the cycle time, the longer the time spent in sleep mode and, therefore, the lower the average energy consumption per cycle.

It can be seen in Figure 14 that, for the STARPORTS use case, with a cycle time of 900 s, sending an average of 32 bytes through LoRa and with an SF of 9, a battery life of 15,720 h (around 1.8 years) is estimated. If we want to extend the battery life up to 3 years, keeping the same range and data size, we should increase the cycle time to 1950 s. In the case of configuring the node with an SF of 12 in order to achieve a higher range, we should increase the cycle time up to 3160 s to get a battery life of 3 years. Otherwise, by using an SF of 7 for short range, we should set the cycle time to 1725 s.

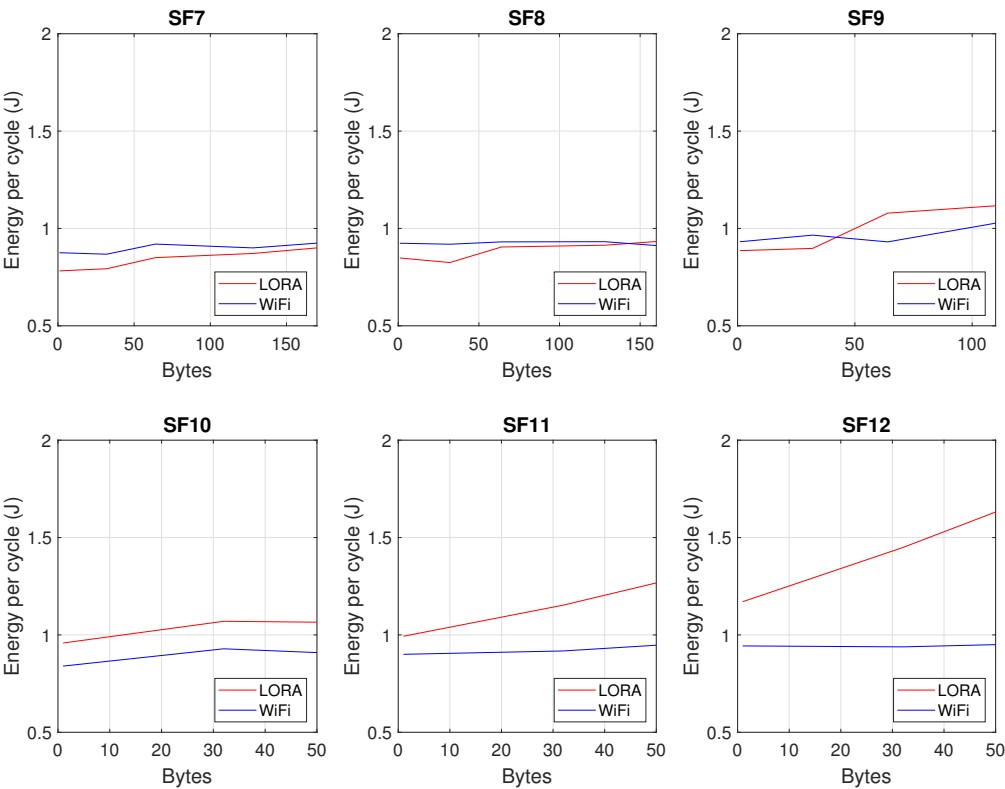

**Figure 12.** Energy consumption by the sensor node for different Spreading Factors (SFs).

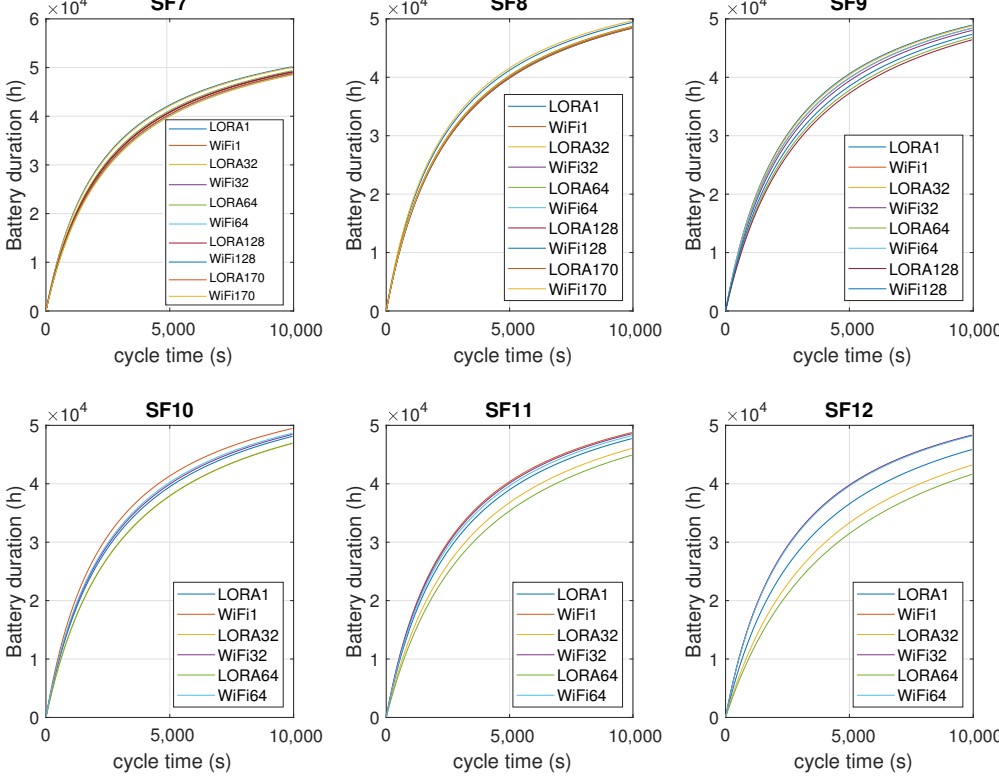

**Figure 13.** Battery duration for different cycle times and SFs.

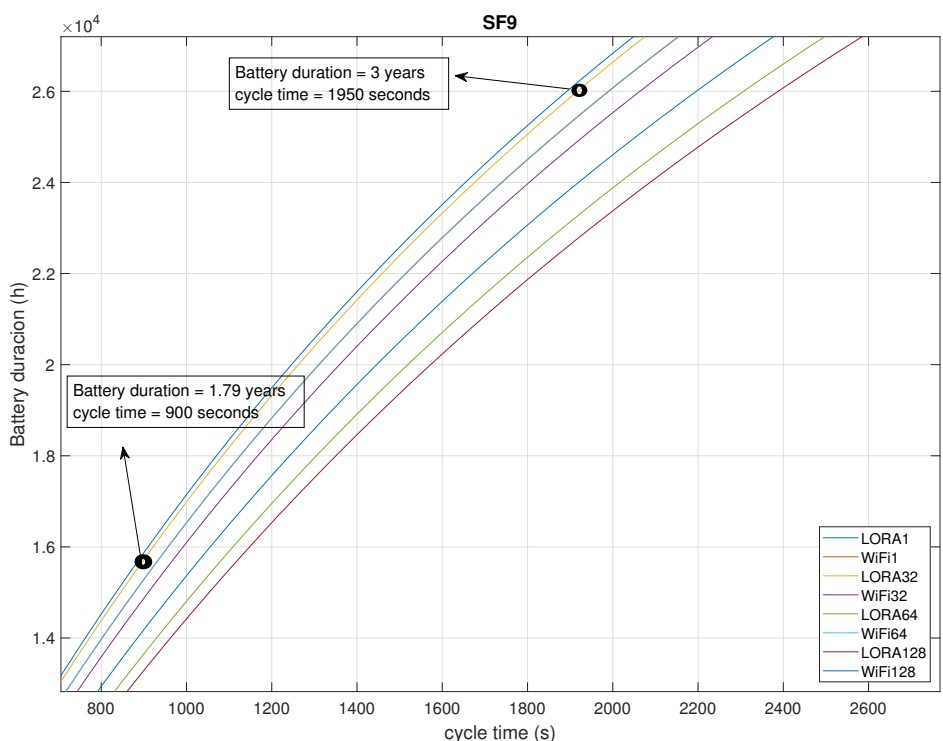

**Figure 14.** Battery duration with the parameters used in the STARPORTS case.

## 6. Conclusions

The new applications that have emerged from the IoT concept have generated a great growth of IoT devices connected to the Internet. This has resulted in a large amount of data flowing into the cloud, which can require a huge bandwidth and in consequence, an inefficient solution from the energy or storage point of view. In addition, certain applications require real-time processing or even make decisions without the need of an Internet connection. In this paper, a flexible Fog Computing architecture has been proposed. Thanks to that flexibility, this approach is capable of solving the previous issues related to the real-time processing and the communications bandwidth, as well as adding new advantages to the existing architectures, such as increasing security and privacy in the network and reducing the required resources at the node level, in order to increase autonomy without losing computational capacity.

In the proposed architecture, low-power strategies have been implemented in the sensor node in order to obtain autonomy for several years. These nodes are capable of selecting between two different communication links (LoRa and WiFi) on the fly, depending on the available coverage, the amount of data to be transmitted, or a trade-off between the energy consumption, coverage, and data to be transmitted. Furthermore, in the proposed architecture, a local server has been implemented within the IoT gateway, in which it is possible to process data at higher speed. This IoT gateway can work in real-time, with a very low latency in response to the data measured by the sensor nodes.

Different tests have been carried out by varying the parameters of the system, such as the duty cycle, the size of the data packet, and the Spreading Factor (SF), which only affects the LoRa communications. The objective of these tests has been to quantify the energy consumption of the sensor nodes focusing on the communication links. We have seen that, unlike WiFi, in the case of LoRa, the energy consumption increases as the number of bytes to be transmitted or the SF is increased. Additionally, there is a cut-off point between the

LoRa and WiFi curves, which, from an energy point of view and depending on the range we need to reach, can help us to determine which technology is more efficient.

The efficiency of the proposed architecture has been tested in a real scenario for a Smart Port application. It has demonstrated the capability of the sensor nodes to capture temperature, corrosion, acceleration, and pressure data and send them to the IoT gateway at a distance of 5.7 km, maintaining, even at that distance, a low energy consumption. Thus, the application has been running for more than one year in a hostile environment. Extrapolating the results obtained from the tests, we estimate that, with a 5800 mA/h battery capacity, these sensor nodes can monitor critical points of the port infrastructure and perform predictive maintenance of structural health for a duration of 1.7 years.

**Author Contributions:** Conceptualization, A.C. and A.I.; Formal analysis, A.I. and M.L.; Funding acquisition, A.I. and A.C.; Investigation, A.C., A.I., M.L., J.C., and A.P.; Methodology, A.I. and M.L.; Project administration, A.I.; Software, M.L., A.I., J.C., and A.P.; Supervision, A.I. and A.C.; Validation, M.L., A.I., and J.C.; Writing—Original draft, M.L.; Writing—Review & editing, A.I., M.L., and A.C. All authors have read and agreed to the published version of the manuscript.

**Funding:** This work was supported by STARPORTS-Sistema Inalámbrico Distribuido de monitorización, prevención y actuación para la Gestión Costera, a FEDER Innterconecta (Canary Island) funded project (ITC-20181029) in collaboration with FCC, Wellness Telecom, SensorLab and PLOCAN.

**Institutional Review Board Statement:** Not applicable.

**Data Availability Statement:** Not applicable.

**Acknowledgments:** This work has been possible thanks to the cooperation of CEIT with FCC, Wellness Telecom, SensorLab, and PLOCAN.

**Conflicts of Interest:** The authors declare no conflict of interest.

## Abbreviations

The following abbreviations are used in this manuscript:

| | |
|---|---|
| AGV | Autonomous Guided Vehicle |
| API | application programming interface |
| ABP | Activation by Personalization |
| CC | Cloudlet Computing |
| DHCP | Dynamic Host configuration Protocol |
| DSP | Digital Signal Processor |
| GPIO | General Purpose Input-Output |
| HW | Hardware |
| IoT | Internet of Things |
| IP | Internet Protocol |
| LoRa | Long Range Modulation |
| LoRaWAN | Long Range Wide Area Network |
| LPWAN | Low Power Wide Area Network |
| MCC | Mobile Cloud Computing |
| MEC | Mobile Edge Computing |
| NPU | Neural Processing Unit |
| OTAA | Over the Air Activation |
| PLOCAN | Plataforma Oceanica de Canarias |
| SSID | Service Set Identifier |
| SW | Software |
| TCP | Transmision Control Protocol |
| UART | Universal Asynchronous Receiver-Transmitter |
| RTC | Real Time Clock |

| SF | Spreading Factor |
|---|---|
| SOM | System on Module |
| UDP | User Datagram Protocol |
| USB | Universal Serial Bus |
| WiFi | Wireless Fidelity |

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
