# Peer review of "A Flexible Fog Computing Design for Low-Power Consumption and Low Latency Applications"

_electronics, doi:10.3390/electronics10010057_

Round 1

Reviewer 1 Report

This paper is investigating a flexible emerging technology namely, fog computing, a model for improving power consumption as well as achieving low latency applications. The paper in good shape. However, the authors must answer the follwoing:

  • The case study of Smart Ports. Is this one the only one you are targeting of there is might be more? IN simple words, the applicability of your system?
  • The system implementation and testing seem nice, however, what about the system complexity, as well as implementation?
  • The model feasibility in terms of the cost?
  • Remove old references. More relevant literature can be considered such:
    • Low-latency vehicular edge: A vehicular infrastructure model for 5G
    • Trustworthy and sustainable smart city services at the edge
    • Reinforcing the edge: Autonomous energy management for mobile device clouds
  • What would be the role of AI in your system?

Reviewer 2 Report

The paper proposes a fog architecture that can use WiFi and LoRa communication technologies. The architecture contains an Internet of Things (IoT) gateway and more sensor nodes. The idea is interesting, but I have the following concerns:

The related work can be extended with a reference architecture for fog computing, like the one proposed by OpenFog Consortium 

The paper should include the general system architecture, from the software and functional point of view.

What is the role of the gateway? Is fog computing done here? what are the services provided by fog computing? Data can be sent to the cloud?

Round 2

Reviewer 1 Report

I have no more comments

Reviewer 2 Report

The paper was improved by the revision process.